# Who Thrives in Medical School? Intrinsic Motivation, Resilience, and Satisfaction Among Medical Students

**DOI:** 10.3390/healthcare13233049

**Published:** 2025-11-25

**Authors:** Julia Terech, Pola Sarnowska, Klaudia Bikowska, Mateusz Guziak, Maciej Walkiewicz

**Affiliations:** 1Division of Quality of Life Research, Department of Psychology, Faculty of Health Sciences with the Institute of Maritime and Tropical Medicine, Medical University of Gdańsk, 80-210 Gdańsk, Poland; julia.terech@gumed.edu.pl (J.T.); klaudia.bikowska@gumed.edu.pl (K.B.); 2Faculty of Medicine, Student Scientific Circle of Psychology, Medical University of Gdańsk, 80-210 Gdańsk, Poland; pola.sarnowska@gumed.edu.pl

**Keywords:** medical education, medical students, satisfaction, motivation, resilience, COVID-19

## Abstract

Background: Medical education is highly demanding and often entails stress, pressure, and competition. Understanding what drives students’ satisfaction is essential to support learning and well-being. This study aims to identify factors associated with satisfaction with medical education among Polish medical students, focusing on motivation, personal circumstances, resilience, and the long-term impact of COVID-19. Methods: In a cross-sectional online survey, 334 students from years one, four, and six completed measures of satisfaction with medical studies (nineteen items), motivation (ten items), resilience (using the Brief Resilience Scale), self-rated health, financial situation, global life satisfaction, and study-related stress, plus eight items on COVID-19 impact. Associations were assessed using Spearman correlations and Mann–Whitney U tests. Results: Higher satisfaction was associated with intrinsic motivation (e.g., personal decision to study medicine or interest in medicine), more favorable personal circumstances (better health, financial situation, higher global life satisfaction, and lower stress), and greater individual resilience. Students reporting pandemic-related setbacks (knowledge gaps, reduced confidence, curtailed clinical exposure, and interpersonal skills) showed lower satisfaction with overall experience, relationships, theoretical and practical classes, and perceived future competence. Conclusions: Intrinsic motivation, resilience, and supportive personal circumstances were linked to higher satisfaction, whereas enduring pandemic disruptions coincided with lower satisfaction across domains. Targeted strategies that cultivate intrinsic motivation and resilience and address financial/health stressors and COVID-19 learning gaps may enhance student satisfaction.

## 1. Introduction

Medical school is demanding and competitive, which can threaten students’ satisfaction with their training [1]. In the Polish context, studies mention discrepancies between students’ expectations and reality, disappointment with teacher support and a lack of focus on practical skills [2], but research directly evaluating medical students’ satisfaction with their course is scarce. Understanding what underpins satisfaction is important because it is linked to engagement and educational outcomes. Intrinsic motives (e.g., interest in medicine, and desire to help others) are reported more frequently than extrinsic ones (e.g., parental expectations, and social status) [3], and may shape students’ satisfaction with their course. Similar findings concern final-year Polish medical students for whom the desire to help others is the most frequently indicated reason to pursue medicine [4]. Greater autonomy over one’s decision to study medicine is linked to better academic and well-being outcomes. Among medical students, intrinsic motivation correlates with higher academic performance and lower study-related exhaustion than extrinsic motivation [5]. This hint at a relationship between medical students’ motivation to study and their course satisfaction could prove beneficial in terms of formulating potential career guidance for prospective medical students, since students with a passion for the subject will likely be more motivated to learn, and grit seems to be associated with medical student performance [6].

Beyond motivation, personal circumstances including life satisfaction and financial strain are linked to how students appraise their education [7,8]. This suggests that financial stability and a lack of anxiety in this regard can be important predictors of student satisfaction with their studies. Medical training is associated with high stress and health strain; increases in depressive symptoms and declines in general health can appear as early as year one and persist thereafter [9,10,11]. Among students studying different medical disciplines in Poland, the self-assessment of health and lifestyle have been found to be positively related to lifestyle practices and satisfaction with life [12]. Institutional factors (assessment and teaching methods) also relate to stress and well-being [13]. Satisfaction is further linked to basic psychological need fulfillment and adaptive mindsets [14]. 

Resilience as the capacity to adapt to adversity has been associated with more effective coping among health professions trainees and may buffer stressors relevant to course satisfaction [15,16,17,18,19]. 

Medical students during the pandemic report feelings of wasted study potential, changes in well-being and ability to learn, and concern about clinical education using e-learning [20]. Many of them believed they should return to clinical rotations [21], some experienced a decline in their study and work performance [22], and lost confidence in their ability to become competent doctors [23]. Interestingly enough, students who attend online courses on history and patient examinations can have higher total knowledge scores than those who attend regular offline classes but, on the other hand, they tend to have lower satisfaction scores [24], which indicates that the relationship between performance and satisfaction both in an online and offline setting may not always be straightforward. Students with a higher resilience score experienced less anxiety and less worry about clinical experience and academic performance [25], and those with lower resilience were more prone to severe burnout and reduced motivation during the pandemic, which emphasizes the need for resilience-building interventions among medical students [26]. There are indications of resilience being associated with feeling capable of helping in a crisis and previous stressful events contributing to building resilience behaviors implemented during the pandemic [27], which highlights the importance of personal experiences in resilience development. Worthy of note is also the fact that the pandemic contributed to the already high prevalence of anxiety and depression among medical students worldwide [28], which may have also affected student response to the COVID crisis.

Few studies have jointly examined motivation, personal circumstances, resilience, and pandemic-related disruptions as correlates of satisfaction with medical education—particularly in Poland. To address this gap, we conducted a cross-sectional study of Polish medical students. We asked the following:

Q1. What is the association between motivational factors (e.g., whether the decision to pursue a medical degree was based on a student’s personal decision, their interests, or whether it was induced externally, to secure a future or social prestige, or due to a desire to feel needed) and satisfaction with medical education?

Q2. How are personal circumstances, like financial situation, health, stress, and general life satisfaction, associated with satisfaction with medical education?

Q3. What is the association between resilience and satisfaction with medical education?

Q4. How does the perceived long-term impact of COVID-19 relate to satisfaction with medical education?

Clarifying these relationships can inform career guidance, curricular adjustments, and student support to foster more satisfied and better prepared physicians [29].

## 2. Materials and Methods

### 2.1. Study Measures

The survey included several questionnaires.

First, the participants answered items addressing sociodemographic factors and selected aspects of medical studies (such as year of study or the participant’s university).

The Brief Resilience Scale [30] was used to describe participants’ mental resilience. The BRS includes six items, each one requiring a response on a five Likert scale. In the current study, authors used a validated Polish version of the BRS, with Cronbach’s alpha coefficient α = 0.88 [31].

Satisfaction with medical studies was measured by a questionnaire (19 items, responses on a seven-Likert scale) developed by the research team based on the prior literature [32,33]. In the post factum validation of the satisfaction questionnaire, the authors obtained Cronbach’s alpha coefficient α = 0.90 and McDonald’s omega reliability coefficient ω = 0.92.

A similar method was used to create a questionnaire of motivation for medical studies (ten items in total, nin of which required a response on a seven-Likert scale and one which was an open-ended question)—authors constructed the questions based on previous Polish research [4,34,35]. In the post factum validation of the motivation questionnaire, the authors obtained Cronbach’s alpha coefficient α = 0.54 and McDonald’s omega reliability coefficient ω = 0.13. Further analysis indicated that internal consistency could be improved; after removing two items, authors obtained Cronbach’s alpha coefficient α = 0.62 and McDonald’s omega reliability coefficient ω = 0.6. Due to such post factum validation results, the nature of the described scale should be considered as exploratory. The questionnaire of motivation aims at distinguishing two major motivational areas that cause students to choose a medical course: intrinsic motivation, such as a person’s autonomous decision, interests, and a desire to feel needed by others; and extrinsic motivation, like other people’s expectations, and a desire for a clear and secure future, social prestige, and knowledge useful in private life.

Authors prepared four items dedicated to self-reported assessment of personal circumstances, which included an evaluation of the participant’s general life satisfaction, as well as its constituents separately, including financial situation, state of health, and stress levels associated with medical studies, both current and in the previous years of study (all rated on a seven-Likert scale).

At the end of the survey, students were asked about the COVID-19 pandemic’s impact on their academic performance. Participants could choose multiple answers from eight items constructed by the authors and based on previous research [20,21,22,23,24].

The full item sets for satisfaction, motivation, personal circumstances, and COVID-19 impact appear in Appendix A. Appendix A also report the post hoc validation results for the satisfaction and motivation questionnaires.

### 2.2. Data Collection

We conducted an online cross-sectional survey using Google Forms. Participants were recruited via social media platforms and medical universities’ internal e-mail boxes. We were gathering data for 3.5 months during the winter semester (including the examination session period). Inclusion criteria were fluency in Polish and current enrollment in the first, fourth, or sixth year of a medical degree program

### 2.3. Participants

A total of 334 Polish medical students (aged 18 to 33; M = 22.26, SD = 2.63) completed the survey; 21% (*n* = 71) were male and 79% (*n =* 263) were female. Most of the participants were from cities up to 100 000 inhabitants (33%, *n* = 109). Students of the first year made up 36% (*n* = 119) of the studied group, 35% (*n* = 118) were fourth-year students, and 29% (*n* = 97) were sixth-year students. The largest institutional subgroups were from the Medical University of Gdansk (35%, *n* = 117) and Poznan University of Medical Sciences (26%, *n* = 87). Full demographic characteristics of participants are presented in Appendix A.

### 2.4. Statistical Analysis

Descriptive statistics were used to describe the participants’ sociodemographic characteristics. The mean score and standard deviation of participants’ age and number of their friends were calculated. Other sociodemographic characteristics were expressed in percentage.

To assess the relationships between students’ motivation to study medicine, an assessment of their personal circumstances, their resilience, and their satisfaction with studying medicine, a Spearman correlation was conducted.

To determine whether satisfaction with studying medicine differed across the groups of students declaring a negative COVID-19 impact on their academic performance and those who did not declare such an impact, a Mann–Whitney U test was performed. The questionnaires’ scores are presented as medians and interquartile ranges.

Since the distribution of participants’ responses departed from normality, we decided to use non-parametric tests. Whenever multiple comparisons occurred, Benjamini–Hochberg *p*-value correction was applied. *p* < 0.05 was considered statistically significant.

### 2.5. Data Quality and Validity

Participation was voluntary, preceded by electronic informed consent, fully anonymous, and conducted under prior ethics committee approval (Bioethics Committee of the Medical University of Gdansk, Poland—NKBBN/235/2023). We used a validated instrument (the Polish version of the Brief Resilience Scale) and report internal consistency for the study-specific scales in the present sample. Prior to analysis, we carried out a response quality screening. In the Discussion, we explicitly address remaining sources of error (self-report and volunteer sampling) and outline implications for the generalizability of the findings.

## 3. Results

### 3.1. Motivation to Study Medicine

A Spearman correlation was conducted to assess the relationship between motivation to study medicine and satisfaction with studies. Numerous significant correlations between various motives for studying and particular components of satisfaction were found, and they are displayed in detail in Table 1.

### 3.2. Assessment of Personal Circumstances

A Spearman correlation was conducted to assess the relationship between assessment of personal circumstances and satisfaction with studies. Numerous significant correlations between assessment of financial situation, health, general life satisfaction, or stress levels and particular components of satisfaction were found, and they are displayed in detail in Table 2.

### 3.3. Individual Resilience

A Spearman correlation was conducted to assess the relationship between individual resilience and satisfaction with studies. Significant correlations between resilience level and particular components of satisfaction were found, and they are displayed in detail in Table 3.

### 3.4. Self-Reported COVID-19 Impact on Students’ Academic Performance

A Mann–Whitney U test was performed to evaluate whether satisfaction with studying medicine differed by the groups of students declaring a negative COVID-19 impact on their academic performance and those who did not declare such an impact (Table 4, Table 5, Table 6 and Table 7).

It was found that students who declare that they have gaps in their knowledge due to the COVID-19 pandemic rate their overall satisfaction with medical studies significantly lower than students who do not declare such concerns, U = 16,706, *p* < 0.001. Furthermore, those with gaps in their knowledge are less satisfied with studying at their current university (*U* = 16,423.5, *p* < 0.001) and their mode of studies (*U* = 14,604, *p* = 0.037), as well as with their knowledge level (*U* = 17,163, *p* < 0.001), practical skills (*U* = 16,303, *p* < 0.001), and relationships with other medical students (*U* = 14,741, *p* = 0.033). They are also significantly less content with theoretical classes (*U* = 15,575.5, *p* = 0.002) and practical classes without patients (*U* = 16,539, *p* < 0.001). More detailed results are displayed in Table 4.

Moreover, the results indicated significant differences in terms of overall satisfaction with medical studies between students who declare they lack interpersonal skills due to the COVID-19 pandemic, and students who do not. The first group is significantly less content with their studies, *U* = 5319.5, *p* = 0.03. They also show lower satisfaction with their relationships with other medical students (*U* = 5353.5, *p* = 0.027), peer relationships outside the medical studies environment (*U* = 5362, *p* = 0.026), and relationships with the medical staff (*U* = 5403.5, *p* = 0.022). More detailed results are displayed in Table 5.

In terms of practical skills, students who declare they lack them due to the COVID-19 pandemic are significantly less satisfied with their medical course in general, *U* = 16,839.5, *p* < 0.001. The results indicated significant differences between this group and students who do not declare such an impact in terms of their satisfaction with the mere fact of being admitted to medical studies (*U* = 14,736, *p* = 0.006), studying at the current university (*U* = 16,546.5, *p* < 0.001), their knowledge level (*U* = 15,822.5, *p* < 0.001), and their practical skills (*U* = 17,283.5, *p* < 0.001). It was also found that students who declare they lack practical skills are significantly less content with their relationships with other medical students (*U* = 14,983, *p* = 0.006), peer relationships outside the medical studies environment (*U* = 14,390, *p* = 0.038), relationships with lecturers (*U* = 14,627.5, *p* = 0.019), theoretical classes (*U* = 15,701.5, *p* < 0.001), and practical classes without patients (*U* = 16,995, *p* < 0.001). More detailed results are displayed in Table 6.

The results showed several differences in satisfaction between students who declare they lack self-confidence as future doctors due to the COVID-19 pandemic and those who do not declare such concerns. The first group is significantly less content with their medical course in general, *U* = 13,655, *p* < 0.001. They are less satisfied with the mere fact of being admitted to medical studies (*U* = 11,347.5, *p* = 0.029) and studying at their current university (*U* = 13,255, *p* < 0.001). Furthermore, it was found that students who lack self-confidence as future doctors present significantly lower satisfaction with their knowledge level (*U* = 13,319.5, *p* < 0.001), practical skills (*U* = 13,436, *p* < 0.001), and the amount of learning material (*U* = 11,439, *p* = 0.04). In terms of social interactions, they are less content with their relationships with other medical students (*U* = 11,922.5, *p* = 0.007), peer relationships outside the medical studies environment (*U* = 12,230, *p* = 0.002), romantic relationships (*U* = 11,618.5, *p* = 0.024), and relationships with lecturers (*U* = 12,427, *p* = 0.001). The results also showed that this group of students is less satisfied with their theoretical classes (*U* = 12,454.5, *p* = 0.001) and practical classes without patients (*U* = 12,952.5, *p* < 0.001). More detailed results are displayed in Table 7.

## 4. Discussion

The study identified factors associated with satisfaction with medical education, allowing for the development of a preliminary profile of students reporting high levels of satisfaction.

A key factor associated with satisfaction is the type of motivation underlying the decision to study medicine. Consistent with Self-Determination Theory, intrinsic (autonomous) motivation is generally positively associated with satisfaction, while extrinsic (controlled) motivation tends to show weaker or null associations [36,37,38]. A notable exception is students’ satisfaction with theoretical courses and practical sessions with patients: in both areas, extrinsic motivation was unrelated to satisfaction, while intrinsic motivation retained a positive correlation. This pattern mirrors evidence that autonomous motivation predicts deeper learning and lower levels of study-related exhaustion among medical students [37,39]. Two psychological mechanisms may be responsible for this pattern. First, to reduce cognitive dissonance, students without strong intrinsic motivation may adopt external justifications for their involvement—for example, viewing theoretical courses as a source of personally useful knowledge, and practical sessions as an investment in a secure professional future and social prestige. Second, the lack of association between extrinsic motivation and course satisfaction may reflect an effort/cost justification effect; participating in time- and effort-intensive activities reduces the likelihood of openly admitting dissatisfaction, as this could negatively impact psychological well-being. Both interpretations are consistent with classic accounts of dissonance and severity-of-initiation effects [40,41]. Overall, intrinsic motivation appears to be the strongest correlate of satisfaction, especially with respect to specific learning experiences. Extrinsic motivation, on the other hand, does not provide comparable benefits. Any apparent “protective” association likely reflects processes of dissonance reduction and cost justification, rather than a true increase in satisfaction. In this regard, educational institutions could focus on providing students with opportunities to foster their intrinsic motivation, such as workshops that prioritize practice, problem-solving, and cooperation over theory presentation and assessment. Student support services, on the other hand, might target the externally induced pressure to become a doctor. 

One of the most often declared motives to study medicine is the desire to help people, as well as having an interest in the medical field [3]. The latter was analyzed in the current study and proved to have a positive relationship with students’ satisfaction with their course. This poses the question of whether intrinsic motives, such as interests, should not be verified along with the candidate’s knowledge during the admissions process at a medical university. For now, there are no such selection methods in Poland. The issue does not seem to be simple to solve, since findings from foreign medical universities, which implemented personal statements or interviews, are unclear [42,43].

The relationship between overall life satisfaction and domain-specific satisfaction has been proven before [7,33]. Therefore, it is not surprising that a significant positive relationship between global life satisfaction and satisfaction with studying medicine was found in the current study. This aligns with ‘spillover’ accounts linking life domain evaluations with global well-being [44] and longitudinal data in medical students [45]. According to the literature, global life satisfaction and domain satisfaction influence one another [7]. Thus, it can be said that a content medical student is the one who initially had high life satisfaction in general. Other global factors influencing an individual’s well-being that were measured in the current study were the assessment of health and financial situation. The positive relationships between them and satisfaction with the medical course were proven; however, they seem to be less significant than in studies conducted previously by other researchers [8,11]. The differences in magnitude compared to those prior cohorts may reflect contextual and institutional factors; nevertheless, links between financial strain, health-related distress, and poorer academic experience are well-documented [46]. Therefore, student support organs should provide interventions tailored to groups of students who are especially at risk of these struggles, such as suitable psychological and financial support, and inclusive facilities.

We examined two workload-related, potentially modifiable, facets of satisfaction: the time spent on their studies and the amount of learning material. Higher satisfaction with the time devoted to medical studies was associated with lower self-reported stress; a parallel association was observed for satisfaction with the amount of curricular content. This accords with reports of perceived curricular overload and associations between sustained fatigue and a poorer educational climate, supporting workload calibration and alignment of assessment with core content [47,48,49,50]. Translating these insights into curricular practice may involve, for example, calibrating workloads to realistic study time and aligning assessment with core content. It may enhance both students’ satisfaction with their program and their psychological well-being.

There are also relationships between satisfaction with specific components of studies and the individual resilience of medical students; the strongest ones regard the amount of learning material and time spent on studies. Medical practitioners’ resilience is shaped by personality traits such as self-directedness and persistence [19], which certainly help to master a large amount of learning material in a short period. Correspondingly, less resilient students experience higher stress levels in the face of material overload, which is associated with lower satisfaction with certain aspects of their studies. In the presented study, individual resilience has a positive relationship with satisfaction with medical studies in various areas, both in learning-related matters and in social interactions. Such a global influence of individual resilience is consistent with the literature on the subject [16]. Prior work in health professions education similarly links resilience to adaptive coping under academic stress [51]. Thus, implementing resilience-building interventions [15] in medical education could benefit future medical practitioners and raise their satisfaction with their studies. We therefore recommend evaluating resilience-building initiatives integrated with curricular demands for feasibility and effectiveness [52]. The current study proves that the long-term impact of the COVID-19 pandemic cannot be ignored when analyzing students’ satisfaction with medical studies. In the literature, students declared that their medical education had been significantly disrupted by the pandemic [21] and their academic performance had deteriorated [20,22], which could result in gaps in theoretical and practical knowledge. Such students should receive appropriate support, since even a few years after the global lockdown they are less content with their medical course and its particular components, such as their knowledge level, practical skills, various types of classes, or peer relationships. Similar relationships were found for students who declare that, due to the pandemic, they lack self-confidence as future doctors, and this should not be ignored either, as such confessions were present in the previous research [23]. Our finding that perceived long-term COVID-19 impacts relate to lower satisfaction coheres with reviews and surveys showing that, although knowledge outcomes were sometimes maintained in online/hybrid formats, satisfaction frequently declined, especially where clinical exposure was restricted [53,54]. During the COVID-19 outbreak, some institutions successfully implemented support programs for health practitioners and students affected by the pandemic’s consequences on their education [55,56], and, as is known, the higher the environmental support and resources, the higher the domain-specific satisfaction [33].

### 4.1. Study Limitations

First, the study was conducted in Poland, which limits the generalizability of the findings to medical students in other cultural contexts, particularly outside Europe [57].

Second, all variables were measured via self-report, which introduces risks of response bias, including social desirability, common method variance, and recall error [58,59].

Third, the cross-sectional design precludes causal inference. A longitudinal design could more effectively capture the dynamics of the examined phenomena and the directionality of associations [60].

### 4.2. Further Research Directions

Our findings can inform medical educators and curriculum planners (e.g., targeting autonomy-supportive teaching and assessment), student support services (screening and tailored support for students at risk due to financial strain, health problems, stress, or low resilience), and career counselors (guidance that considers the motivational profile of prospective and current medical students).

Future studies should address the present limitations. First, to improve generalizability, multi-site, cross-cultural replications using harmonized measures are needed. Second, to reduce self-report bias, mixed-methods designs could combine validated questionnaires with behavioral or institutional indicators (e.g., attendance, progression, and exam performance) and qualitative interviews. Third, to strengthen causal inference, longitudinal designs (e.g., cohort studies and cross-lagged panel models) and intervention trials should test whether changes in motivation, resilience, or stress predicts subsequent changes in satisfaction. Finally, future work could evaluate implementation strategies, such as autonomy-supportive teaching workshops, resilience-building programs, and financial counseling, and report their effectiveness.

## 5. Conclusions

Among the surveyed medical students in Poland, higher satisfaction with medical education was associated with intrinsic, self-initiated motivation, particularly with a genuine interest in medicine. Within this sample, more favorable personal circumstances—such as greater life satisfaction, better perceived health, and more stable financial situation—together with lower levels of study-related stress (notably where workload and time demands were perceived as manageable) were also linked to higher satisfaction. Greater individual resilience was positively associated as well. Conversely, students who reported enduring COVID-19-related disruptions, perceived knowledge gaps, curtailed practical or interpersonal skills, and reduced confidence in their future competence tended to report lower satisfaction levels. These associations should be interpreted within the specific academic and institutional context of the participants.

## Figures and Tables

**Table 1 healthcare-13-03049-t001:** Correlation coefficients between motives to study medicine and specific components of satisfaction with studies.

Satisfaction Component	Motivation to Study Medicine						
	My Conscious Decision	A Decision Made due to Others’ Expectations	A Clear and Secure Future	My Interests	Social Prestige	Knowledge Useful in Private Life	A Desire to Feel Needed by Others
Overall satisfaction with medical studies	0.33 ***	−0.28 ***		0.43 ***			
The mere fact of being admitted to medical studies	0.29 ***	−0.26 ***	0.15 *	0.38 ***	0.17 **	0.2 **	0.23 ***
Studying at your current university	0.31 ***	−0.29 ***		0.36 ***			
Mode of studies: full-time/part-time	0.23 ***	−0.18 **		0.19 **			
My knowledge level	0.27 ***	−0.23 ***		0.32 ***			
My practical skills	0.31 ***	−0.24 ***		0.3 ***			
The amount of learning material	0.21 ***	−0.2 **		0.29 ***			
Time spent on studies	0.21 ***	−0.16 *		0.26 ***			
Relationships with other medical students	0.2 **	−0.15 *					
Peer relationships (outside the medical studies environment)	0.21 ***	−0.16 *		0.14 *			
Romantic relationships			0.13 *	0.15 *		0.15 *	
Relationships with patients				0.16 *			
Relationships with lecturers	0.21 ***	−0.22 ***		0.28 ***		0.16 *	
Relationships with the medical staff (apart from those who hold classes)				0.17**		0.13 *	
Theoretical classes	0.18 **			0.25 ***		0.14 *	
Practical classes without patients	0.13 *	−0.13 *		0.25 ***			
Practical classes with patients	0.16 *		0.14 *	0.26 ***	0.16 *		
Student internships				0.18 **			

* *p* < 0.05. ** *p* < 0.01. *** *p* < 0.001.

**Table 2 healthcare-13-03049-t002:** Correlation coefficients between the assessment of personal circumstances and specific components of satisfaction with studies.

Satisfaction Component	Assessment of Personal Circumstances					
	Assessment of Financial Situation	Assessment of Health	Assessment of General Life Satisfaction	Assessment of Current Stress Level	Assessment of Stress Level in Previous Years	
					Fourth Year Students	Sixth Year Students
Overall satisfaction with medical studies	0.13 *	0.18 **	0.44 ***	0.29 ***	0.36 ***	0.36 ***
The mere fact of being admitted to medical studies		0.21 ***	0.25 ***			
Studying at your current university	0.14 *	0.23 ***	0.35 ***	0.26 ***	0.3 **	0.24 *
Mode of studies: full-time/part-time	0.12 *	0.22 ***	0.2 ***	0.18**		
My knowledge level	0.18 **	0.29 ***	0.34 ***	0.33 ***	0.28 **	0.4 ***
My practical skills			0.28 ***	0.24 ***		0.44 ***
Extra activity during studies (e.g., scientific, social, artistic, sports)	0.14 *	0.13 *	0.27 ***	0.23 ***		0.26*
The amount of learning material	0.25 ***	0.29 ***	0.4 ***	0.46 ***	0.31**	0.31 **
Time spent on studies	0.24 ***	0.27 ***	0.44 ***	0.55 ***	0.35 ***	0.36 ***
Relationships with other medical students	0.17 **	0.25 ***	0.33 ***	0.29 ***	0.3 **	0.35 **
Peer relationships (outside the medical studies environment)	0.2 ***	0.27 ***	0.4 ***	0.3 ***	0.23 *	0.44 ***
Romantic relationships	0.2 ***	0.12*	0.27 ***			
Relationships with patients		0.21 ***	0.2 ***	0.11 *	0.23 *	0.25 *
Relationships with lecturers	0.23 ***	0.25 ***	0.36 ***	0.3 ***	0.28**	0.34**
Relationships with the medical staff (apart from those who hold classes)	0.13 *	0.17 **	0.23 ***	0.11*		0.31**
Theoretical classes	0.15 **	0.22 ***	0.3 ***	0.22 ***		0.39 ***
Practical classes without patients	0.12 *	0.15 **	0.26 ***	0.17 **	0.26 **	0.24 *
Practical classes with patients		0.15*	0.19 ***	0.15 **		0.35 **
Student internships	0.11 *		0.12 *	0.11 *		

* *p* < 0.05. ** *p* < 0.01. *** *p* < 0.001.

**Table 3 healthcare-13-03049-t003:** Correlation coefficients between individual resilience and specific components of satisfaction with studies.

Satisfaction Component	Individual Resilience
Overall satisfaction with medical studies	0.27 ***
Studying at your current university	0.18 **
My knowledge level	0.24 ***
My practical skills	0.14 *
Extra activity during studies (e.g., scientific, social, artistic, sports)	0.13 *
The amount of learning material	0.29 ***
Time spent on studies	0.27 ***
Relationships with other medical students	0.24 ***
Peer relationships (outside the medical studies environment)	0.25 ***
Romantic relationships	0.16 **
Relationships with patients	0.15 *
Relationships with lecturers	0.2 ***

* *p* < 0.05. ** *p* < 0.01. *** *p* < 0.001.

**Table 4 healthcare-13-03049-t004:** Differences in satisfaction with studies depending on declared gaps in knowledge.

Satisfaction Component	Median (IQR)		*U*	*p*
	Students Who Declare They Have Gaps in Knowledge (*n* = 117)	Students Who Do Not Declare They Have Gaps in Knowledge (*n* = 217)		
Overall satisfaction with medical studies	5 (3–6)	6 (5–6)	16,706	0.000 ***
Studying at your current university	5 (4–6)	6 (5–7)	16,423.5	0.000 ***
Mode of studies: full-time/part-time	6 (5–7)	7 (6–7)	14,604	0.037 *
My knowledge level	3 (3–5)	5 (4–6)	17,163	0.000 ***
My practical skills	3 (2–5)	4 (2–5)	16,303	0.000 ***
Relationships with other medical students	5 (3–6)	5 (4–6)	14,741	0.033 *
Theoretical classes	4 (3–5)	5 (3–6)	15,575.5	0.002 **
Practical classes without patients	4 (3–5)	5 (4–6)	16,539	0.000 ***

* *p* < 0.05. ** *p* < 0.01. *** *p* < 0.001.

**Table 5 healthcare-13-03049-t005:** Differences in satisfaction with studies depending on lack of interpersonal skills.

Satisfaction Component	Median (IQR)		*U*	*p*
	Students Who Declare They Lack Interpersonal Skills (*n* = 27)	Students Who Do Not Declare They Lack Interpersonal Skills (*n* = 307)		
Overall satisfaction with medical studies	5 (2–5.5)	5 (4.5–6)	5319.5	0.03 *
Relationships with other medical students	4 (3–5)	5 (3–6)	5353.5	0.027 *
Peer relationships (outside the medical studies environment)	5 (3–6)	6 (5–7)	5362	0.026 *
Relationships with the medical staff (apart from those who hold classes)	4 (4–5)	5 (4–7)	5403.5	0.022 *

* *p* < 0.05

**Table 6 healthcare-13-03049-t006:** Differences in satisfaction with studies depending on lack of practical skills.

Satisfaction Component	Median (IQR)		*U*	*p*
	Students Who Declare They Lack Practical Skills (*n* = 112)	Students Who Do Not Declare They Lack Practical Skills (*n* = 222)		
Overall satisfaction with medical studies	5 (2–6)	6 (5–6)	16,839.5	0.000 ***
The mere fact of being admitted to medical studies	6 (6–7)	7 (6–7)	14,736	0.006 **
Studying at your current university	5 (3.75–6)	6 (5–7)	16,546.5	0.000 ***
My knowledge level	5 (2.75–5)	5 (3.25–6)	15,822.5	0.000 ***
My practical skills	3 (1.75–5)	5 (3–6)	17,283.5	0.000 ***
Relationships with other medical students	5 (3–6)	6 (4–6)	14,983	0.006 **
Peer relationships (outside the medical studies environment)	5 (4–6)	6 (5–7)	14,390	0.038 *
Relationships with lecturers	5 (4–6)	5 (4.25–6)	14,627.5	0.019 *
Theoretical classes	4 (3–5)	5 (3–6)	15,701.5	0.000 ***
Practical classes without patients	4 (3–5)	5 (4–6)	16,995	0.000 ***

* *p* < 0.05. ** *p* < 0.01. *** *p* < 0.001.

**Table 7 healthcare-13-03049-t007:** Differences in satisfaction with studies depending on students’ lack of self-confidence as future doctors.

Satisfaction Component	Median (IQR)		*U*	*p*
	Students Who Declare They Lack Self-Confidence as Future Doctors (*n* = 75)	Students Who Do Not Declare They Lack Self-Confidence as Future Doctors (*n* = 259)		
Overall satisfaction with medical studies	**5 (2–5.5)**	6 (5–6)	13,655	0.000 ***
The mere fact of being admitted to medical studies	6 (6–7)	7 (6–7)	11,347.5	0.029 *
Studying at your current university	5 (3–6)	6 (5–7)	13,255	0.000 ***
My knowledge level	3 (2–5)	5 (3–6)	13,319.5	0.000 ***
My practical skills	2 (1–5)	4 (2–5)	13,436	0.000 ***
The amount of learning material	3 (2–5)	4 (2–5)	11,439	0.04 *
Relationships with other medical students	4 (3–6)	5 (3.5–6)	11,922.5	0.007 **
Peer relationships (outside the medical studies environment)	5 (4–6)	6 (5–7)	12,230	0.002 **
Romantic relationships	4 (2–6.5)	6 (3–7)	11,618.5	0.024 *
Relationships with lecturers	5 (3–6)	5 (4.5–6)	12,427	0.001 ***
Theoretical classes	3 (2–5)	5 (3–6)	12,454.5	0.001 ***
Practical classes without patients	4 (3–5)	5 (4–6)	12,952.5	0.000 ***

* *p* < 0.05. ** *p* < 0.01. *** *p* < 0.001.

## Data Availability

The dataset includes sensitive information related to participants’ psychological characteristics and self-reported experiences. In accordance with ethical standards and the conditions approved by the institutional ethics committee, the data cannot be shared publicly to protect participant confidentiality. De-identified data are available upon reasonable request from the corresponding author.

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
