# Peer review of "Who Thrives in Medical School? Intrinsic Motivation, Resilience, and Satisfaction Among Medical Students"

_healthcare, 2025, doi:10.3390/healthcare13233049_

Round 1
Reviewer 1 Report
Comments and Suggestions for Authors
Dear authors,
Your study is important and timely. In your study you addressed an under-researched population (Polish medical students) and you combined motivational and contextual variables together.
My major concern with your study is the measurement validity. Some of your measurement instruments were self-developed and have low internal consistency (for example: the motivation questionnaire, α = 0.54–0.62; ω = 0.13–0.60). These low coefficients limit the study’s interpretability. You should either write that such scales were exploratory and explicitly label analyses as preliminary, or you could consider revising or replacing them with validated instruments.
My second suggestion would be about the construct clarity. The boundaries between “intrinsic/extrinsic motivation” and “personal circumstances” are sometimes confusing. If you write a clearer theoretical framework or draw a figure which would help readers follow the conceptual logic will be appropriate.
My third suggestion will be about the discussion section. You could discuss your findings more on the implications for curriculum design or student-support interventions.
Comments on the Quality of English LanguagePlease check English phrasing for conciseness. Several sentences in the Introduction can be shortened.
Author Response
Dear Reviewer,
We sincerely thank you for your thoughtful and encouraging feedback. We are pleased that you found our study both important and timely, and we greatly appreciate your recognition that it addresses an under-researched population of Polish medical students. We are also grateful for your positive assessment of our approach, particularly the combination of motivational and contextual variables, which we likewise consider a key strength of the manuscript.
Sincerely,
Comment 1: My major concern with your study is the measurement validity. Some of your measurement instruments were self-developed and have low internal consistency (for example: the motivation questionnaire, α = 0.54–0.62; ω = 0.13–0.60). These low coefficients limit the study’s interpretability. You should either write that such scales were exploratory and explicitly label analyses as preliminary, or you could consider revising or replacing them with validated instruments.
Response 1: Thank you for this valuable comment. In the Materials and Methods section, we emphasized the exploratory nature of the motivation questionnaire.
Comment 2: My second suggestion would be about the construct clarity. The boundaries between “intrinsic/extrinsic motivation” and “personal circumstances” are sometimes confusing. If you write a clearer theoretical framework or draw a figure which would help readers follow the conceptual logic will be appropriate.
Response 2: Thank you for this valuable suggestion to improve the clarity of our methodology. In section Materials and Methods, we refined the theoretical framework and described in detail the intrinsic and extrinsic motivators measured by the questionnaire of motivation, as well as the exact components of the personal circumstances that we refer to.
Comment 3: My third suggestion will be about the discussion section. You could discuss your findings more on the implications for curriculum design or student-support interventions.
Response 3: Thank you for drawing our attention to focusing on the practical relevance of our research. In the Discussion section, we outlined curricular suggestions that are relevant to the outcomes of our study.

Reviewer 2 Report
Comments and Suggestions for Authors
Dear Authors(s),
I think that your manuscript is coherent, clearly written and reflects today s Medical world. While peer-reviewing it, I noted down a few improvement suggestions.
Kindly, see below the improvement suggestions:
- The Introduction section would benefit from an information about the Medical students satisfaction in Poland. If this information is available, it would be appropriate to put the research in a context, namely some characteristics of the Medical students in Poland;
- The Material and methods section definitely needs a Descriptive statistics section in which to include the information related to statistical tests and for what they were used. A statement should refer to the motivation of selecting non-parametric tests (Mann-Whitney and Spearman). I am referring here to the normality tests, Shapiro-Wilk and their outcomes. Also, please write what you have used for the quantitative variables (the median and the IQ);
- The Discussion section may be improved by making the connection between the findings and the literature more than it is specified in the manuscript.
- Please extend the limitations subsection by citing references
- There should also be a further research directions subsection in which you can include information about who and how can benefit from the findings of the manuscript. Some further research directions may be answers to the limitations.
Thank you and good luck!
Author Response
Dear Reviewer,
We thank you for your positive assessment of the coherence and clarity of our manuscript and for recognizing its relevance to contemporary medical education. We are very grateful for the time and effort you devoted to reviewing our work, as well as for your valuable and constructive suggestions. We confirm that each of your comments has been carefully considered and addressed in the revised version of the manuscript. Our point-by-point responses are presented below.
Sincerely,
Comment 1: The Introduction section would benefit from an information about the Medical students satisfaction in Poland. If this information is available, it would be appropriate to put the research in a context, namely some characteristics of the Medical students in Poland;
Response 1: Thank you for this suggestion. In the Introduction section, we added information about Polish medical students in the areas of satisfaction, motivation and personal circumstances based on a literature review.
Comment 2: The Material and methods section definitely needs a Descriptive statistics section in which to include the information related to statistical tests and for what they were used. A statement should refer to the motivation of selecting non-parametric tests (Mann-Whitney and Spearman). I am referring here to the normality tests, Shapiro-Wilk and their outcomes. Also, please write what you have used for the quantitative variables (the median and the IQ);
Response 2: Thank you for such an essential insight. To enhance methodological transparency, the Materials and Methods section was supplemented with a Statistical Analysis subsection, offering a detailed description and rationale for the statistical procedures applied to the collected data.
Comment 3: The Discussion section may be improved by making the connection between the findings and the literature more than it is specified in the manuscript.
Response 3: Thank you for this constructive suggestion. In the revised Discussion, we strengthened the linkage between our findings and prior literature.
Comment 4: Please extend the limitations subsection by citing references.
Response 4: Thank you for this suggestion. We have expanded the Study limitations subsection.
Comment 5: There should also be a further research directions subsection in which you can include information about who and how can benefit from the findings of the manuscript. Some further research directions may be answers to the limitations.
Response 5: Thank you for this helpful suggestion. We have added a “Further research directions” subsection. It outlines who can benefit from our findings (medical educators, curriculum planners, student support services, and career counsellors) and specifies how future studies could address current limitations. The subsection proposes multi-site and cross-cultural replications, longitudinal and experimental designs, and mixed-methods approaches linking self-reports to objective indicators.

Reviewer 3 Report
Comments and Suggestions for Authors
The manuscript addresses an interesting and relevant topic concerning satisfaction and resilience among medical students. This issue has gained particular importance after the COVID-19 pandemic, during which resilience levels in students have reportedly declined. However, several aspects should be addressed before the manuscript can be considered for publication.
-
Data Collection Methodology:
The study mentions the use of an online survey; however, it is unclear how the authors ensured the validity and reliability of the data collected. The manuscript should specify how the researchers guaranteed that the responses were truthful and that there was no fabrication or bias from participants. -
Comparison Between Student Cohorts:
The authors compare first-, fourth-, and sixth-year medical students. Naturally, there are inherent differences between newly enrolled students and those in advanced years of study. The manuscript should explain how the authors accounted for these differences to ensure that comparisons across cohorts are valid and meaningful. -
COVID-19 Impact and Literature Support:
Although the manuscript mentions the impact of COVID-19 on students’ resilience, the introduction lacks sufficient depth and supporting literature to substantiate this statement. The authors should provide a more comprehensive review of previous studies examining how the pandemic affected medical students’ resilience, and consider whether related factors such as anxiety or depression may have influenced their findings. -
Ethical Considerations and Study Context:
The Materials and Methods section lacks details regarding ethical approval. The authors must clearly state whether the study was approved by an ethics committee and describe how informed consent was obtained. Additionally, they should indicate when and where data were collected, specifying whether it was before or after examination periods, as these conditions may significantly influence students’ responses. -
Overgeneralization of Conclusions:
The conclusions appear overstated and should be more tightly aligned with the actual data obtained. For example, the authors state that higher satisfaction with medical education is associated with personal motivation and interest in medicine. Such statements should be contextualized within the specific population studied (i.e., by academic year and institution) to avoid overgeneralization. -
Formatting and Referencing Issues:
The reference list does not comply with the journal’s formatting standards. The in-text citations and reference style, particularly in the Materials and Methods section, should be revised according to the Healthcare guidelines. -
Tables and Statistical Significance:
In Tables 2 through 6, asterisks are used to denote p-values; however, the table legends do not clarify the significance thresholds or the meaning of the symbols. These details should be added to ensure clarity. -
Structure and Highlights Section:
The “Highlights” section is not required by the Healthcare journal and should be removed. Furthermore, the introduction is excessively long and should be condensed to focus more concisely on the study rationale and objectives.
In summary, while the topic is of interest and potential relevance to the journal’s readership, the manuscript requires substantial methodological clarification, ethical transparency, and editorial revision before it can be recommended for publication. I hope that these comments will assist the authors in improving the quality and clarity of their work. Good luck¡¡¡
Comments on the Quality of English LanguageIn addition, the English language requires careful revision to improve clarity, grammar, and fluency. The authors are encouraged to have the manuscript reviewed by a native English speaker or a professional language-editing service prior to resubmission.
Author Response
Dear Reviewer,
We would like to sincerely thank you for your positive and encouraging remarks. We are grateful for your recognition of the relevance and timeliness of our work, particularly in the context of the impact of the COVID-19 pandemic on medical students. We have carefully considered all subsequent comments and have thoroughly addressed each of the specific points raised to improve the clarity, robustness, and overall quality of the manuscript. Our detailed, point-by-point responses are provided below.
Sincerely,
Comment 1: Data Collection Methodology:
The study mentions the use of an online survey; however, it is unclear how the authors ensured the validity and reliability of the data collected. The manuscript should specify how the researchers guaranteed that the responses were truthful and that there was no fabrication or bias from participants.
Response 1: We agree with the Reviewer that it was not possible to “guarantee” fully reliable responses at the level of an individual respondent. Our aim, therefore, was to minimize the risk of fabricated or careless responding as much as possible and to document data quality transparently. In the manuscript, we added a dedicated subsection, Data quality and validity in the Materials and Methods section, where we specify that participation was voluntary, preceded by electronic informed consent, fully anonymous, and conducted under ethics committee approval. We used a validated instrument (the Polish version of the Brief Resilience Scale) and reported reliability for the study-specific scales in our sample. Before the analysis, we carried out a response quality screening. In the Discussion, we addressed remaining sources of error (self-report, volunteer sampling) and indicated the implications for the generalizability of the findings. We acknowledge that it was not possible to eliminate the risk of untruthful responses. Nevertheless, we applied established design and control procedures and presented empirical indicators of measurement quality. We sincerely hope that these efforts will reassure the Reviewer of the credibility of our conclusions within the studied population.
Comment 2: Comparison Between Student Cohorts:
The authors compare first-, fourth-, and sixth-year medical students. Naturally, there are inherent differences between newly enrolled students and those in advanced years of study. The manuscript should explain how the authors accounted for these differences to ensure that comparisons across cohorts are valid and meaningful.
Response 2: We appreciate the Reviewer’s insightful comment. Indeed, we acknowledge that first-, fourth-, and sixth-year medical students differ in experience, workload, and adaptation to medical school demands. Our primary aim, however, was not to perform direct statistical comparisons between cohorts but to explore general associations between motivational, personal, and contextual factors and satisfaction with medical education within a diverse sample representing different stages of training. Including students from these three key points of medical education allowed us to capture a broad range of experiences- from early adjustment to pre-graduation reflection- thereby enhancing the ecological validity of the findings. Nevertheless, we took steps to account for cohort-related heterogeneity. Specifically, majority of analyses were performed across the full sample rather than by direct between-year comparison, and non-parametric methods (Spearman correlations, Mann–Whitney U tests) were used to account for violations of normality and potential distributional differences between groups.
Comment 3: COVID-19 Impact and Literature Support:
Although the manuscript mentions the impact of COVID-19 on students’ resilience, the introduction lacks sufficient depth and supporting literature to substantiate this statement. The authors should provide a more comprehensive review of previous studies examining how the pandemic affected medical students’ resilience, and consider whether related factors such as anxiety or depression may have influenced their findings.
Response 3: Thank you for this insight. In the Introduction section, we strengthened the link between the COVID-19 pandemic and medical student resilience by once again reviewing appropriate literature.
Comment 4: Ethical Considerations and Study Context:
The Materials and Methods section lacks details regarding ethical approval. The authors must clearly state whether the study was approved by an ethics committee and describe how informed consent was obtained. Additionally, they should indicate when and where data were collected, specifying whether it was before or after examination periods, as these conditions may significantly influence students’ responses.
Response 4: Thank you for such an essential insight. To clarify the ethical basis of the study, we supplemented the Materials and Methods section with a Data Quality and Validity subsection. Furthermore, as we recognize the importance of the situational context of gathering data for the study, we expanded the Data Collection subsection with additional information.
Comment 5: Overgeneralization of Conclusions:
The conclusions appear overstated and should be more tightly aligned with the actual data obtained. For example, the authors state that higher satisfaction with medical education is associated with personal motivation and interest in medicine. Such statements should be contextualized within the specific population studied (i.e., by academic year and institution) to avoid overgeneralization.
Response 5: We thank the reviewer for this valuable observation. We agree that the original conclusion could have implied broader generalizability than warranted by our data. In response, we have revised the Conclusions section to more explicitly situate our findings within the context of the study sample. The updated version now specifies that the associations observed refer to the surveyed students in Poland and emphasizes that these results should be interpreted within this specific academic and institutional setting.
Comment 6:. Formatting and Referencing Issues:
The reference list does not comply with the journal’s formatting standards. The in-text citations and reference style, particularly in the Materials and Methods section, should be revised according to the Healthcare guidelines.
Response 6: Thank you for pointing this out. We have revised the in-text citations and reference list to fully comply with the Healthcare journal guidelines.
Comment 7: Tables and Statistical Significance:
In Tables 2 through 6, asterisks are used to denote p-values; however, the table legends do not clarify the significance thresholds or the meaning of the symbols. These details should be added to ensure clarity.
Response 7: Thank you for this valuable comment. The tables have been supplemented with the legends regarding p-values.
Comment 8: Structure and Highlights Section:
The “Highlights” section is not required by the Healthcare journal and should be removed. Furthermore, the introduction is excessively long and should be condensed to focus more concisely on the study rationale and objectives.
Response 8: Thank you for this comment. We have removed the “Highlights” section in the revised version. In the revised manuscript we condensed the Introduction, removing tangential content and repetitions, and focused more explicitly on the study rationale and objectives. Background on resilience and the pandemic was reduced to 1–2 sentences (with details moved to the Supplementary Materials where appropriate). The final paragraph now clearly states the four research questions

Round 2
Reviewer 1 Report
Comments and Suggestions for Authors
Dear authors,
All my concerns were answered.
Thank you.
Comments on the Quality of English LanguagePlease check English phrasing for conciseness. Several sentences in the Introduction can be shortened.
Reviewer 3 Report
Comments and Suggestions for Authors
The manuscript could be accepted for publication.
Comments on the Quality of English LanguageSome grammar mistakes.